# Differentially Private Bayesian Inference for Exponential Families

**Garrett Bernstein**
College of Information and Computer Sciences
University of Massachusetts Amherst
Amherst, MA 01002
gbernstein@cs.umass.edu

**Daniel Sheldon**
College of Information and Computer Sciences
University of Massachusetts Amherst
Amherst, MA 01002
sheldon@cs.umass.edu

## Abstract

The study of private inference has been sparked by growing concern regarding the analysis of data when it stems from sensitive sources. We present the first method for private Bayesian inference in exponential families that properly accounts for noise introduced by the privacy mechanism. It is efficient because it works only with sufficient statistics and not individual data. Unlike other methods, it gives properly calibrated posterior beliefs in the non-asymptotic data regime.

## 1   Introduction

Differential privacy is the dominant standard for privacy [1]. A randomized algorithm that satisfies differential privacy offers protection to individuals by guaranteeing that its output is insensitive to changes caused by the data of any single individual entering or leaving the data set. An algorithm can be made differentially private by applying one of several general-purpose mechanisms to randomize the computation in an appropriate way, for example, by adding noise calibrated to the *sensitivity* of the quantity being computed, where sensitivity captures how much the quantity depends on any individual's data [1]. Due to the obvious importance of protecting individual privacy while drawing population level inferences from data, differentially private algorithms have been developed for a broad range of machine learning tasks [2–9].

There is a growing interest in private methods for Bayesian inference [10–14]. In Bayesian inference, a modeler selects a prior distribution $p(\theta)$ over some parameter, observes data $x$ that depends probabilistically on $\theta$ through a model $p(x \mid \theta)$, and then reasons about $\theta$ through the posterior distribution $p(\theta \mid x)$, which quantifies updated beliefs and uncertainty about $\theta$ after observing $x$. Bayesian inference is a core machine learning task and there is an obvious need to be able to conduct it in a way that protects privacy when $x$ is sensitive. Additionally, recent work has identified surprising connections between sampling from posterior distributions and differential privacy—for example, a single perfect sample from $p(\theta \mid x)$ satisfies differential privacy for some setting of the privacy parameter [10–13].

An "obvious" way to conduct private Bayesian inference is to privatize the computation of the posterior, that is, to design a differentially private algorithm $\mathcal{A}$ that outputs $y = \mathcal{A}(x)$ with the goal that $y \approx p(\theta \mid x)$ is a privatized representation of the posterior. However, using $y$ directly as "the

posterior" will not correctly quantify beliefs, because the Bayesian modeler never observes $x$, she observes $y$; her posterior beliefs are now quantified by $p(\theta \mid y)$.

This paper will take a different approach to private Bayesian inference by designing a pair of algorithms: The release mechanism $\mathcal{A}$ computes a private statistic $y = \mathcal{A}(x)$ of the input data; the inference algorithm $\mathcal{P}$ computes $p(\theta \mid y)$. These algorithms should satisfy the following criteria:

- **Privacy**. The release mechanism $\mathcal{A}$ is differentially private. By the post-processing property of differential privacy [15], all further computations are also private.
- **Calibration**. The inference algorithm $\mathcal{P}$ can efficiently compute or approximate the correct posterior, $p(\theta \mid y)$ (see Section 4 for our process to measure calibration).
- **Utility**. Informally, the statistic $y$ should capture "as much information as possible" about $x$ so that $p(\theta \mid y)$ is "close" to $p(\theta \mid x)$ (see Section 4 for our process to measure utility).

Importantly, the release mechanism $\mathcal{A}$ is public, so the distribution $p(y \mid x)$ is known. Williams and McSherry first suggested conducting inference on the output of a differentially private algorithm and showed how to do this for the factored exponential mechanism [16]; see also [17–20].

Our work focuses specifically on Bayesian inference when the private data $X = x_{1:n}$ is an iid sample of (publicly known) size $n$ from an exponential family model $p(x_i \mid \theta)$. Exponential families include many of the most familiar parametric probability models. We will adopt a straightforward release mechanism where the Laplace mechanism [1] is used to release noisy sufficient statistics $y$ [12, 19], which are a finite-dimensional quantity that capture all the information about $\theta$ [21].

The technical challenge is then to develop an efficient general-purpose inference algorithm $\mathcal{P}$. One challenge is computational efficiency. The exact posterior $p(\theta \mid y) \propto \int p(\theta)p(x_{1:n} \mid \theta)p(y|x_{1:n})dx_{1:n}$ integrates over all possible data sets [16], which is intractable to do directly for large $n$. We integrate instead over the sufficient statistics $s$, which have fixed dimension and completely characterize the posterior; furthermore, since they are a sum over individuals, $p(s \mid \theta)$ is asymptotically normal. We develop an efficient Gibbs sampler that uses a normal approximation for $s$ together with variable augmentation to model the Laplace noise in a way that yields simple updates [22].

A second challenge is that the sufficient statistics may be unbounded, which makes their release incompatible with the Laplace mechanism. We address this by imposing truncation bounds and only computing statistics from data that fall within the bounds. We show how to use automatic differentiation and a "random sum" central limit theorem to compute the parameters of the normal approximation $p(s \mid \theta)$ for a *truncated* exponential family when the number of individuals that fall within the truncation bounds is unknown.

Our overall contribution is the pairing of an existing simple release mechanism $\mathcal{A}$ with a novel, efficient, and general-purpose Gibbs sampler $\mathcal{P}$ that meets the criteria outlined above for private Bayesian inference in any univariate exponential family or multivariate exponential family with bounded sufficient statistics.[1] We show empirically that when compared with competing methods, ours is the only one that provides properly calibrated beliefs about $\theta$ in the non-asymptotic regime, and that it provides good utility compared with other private Bayesian inference approaches.

## 2 Differential Privacy

Differential privacy requires that an individual's data has a limited effect on the algorithm's behavior. In our setting, a data set $X = x_{1:n} := (x_1, \ldots, x_n)$ consists of records from $n$ individuals, where $x_i \in \mathbb{R}^d$ is the data of the $i$th individual. We will assume $n$ is known. Differential privacy reasons about the hypothesis that one individual chooses not to remove their data from the data set, and their record is replaced by another one.[2] Let nbrs$(X)$ denote the set of data sets that differ from $X$ by exactly one record—i.e., if $X' \in$ nbrs$(X)$, then $X' = (x_{1:i}, x_i', x_{i+1:n})$ for some $i$.

**Definition 1** (Differential Privacy; Dwork et al. [1]). *A randomized algorithm $\mathcal{A}$ satisfies $\epsilon$-differential privacy if for any input $X$, any $X' \in nbrs(X)$ and any subset of outputs $O \subseteq Range(\mathcal{A})$,*
$$\Pr[\mathcal{A}(X) \in O] \leq \exp(\epsilon) \Pr[\mathcal{A}(X') \in O].$$

We achieve differential privacy by injecting noise into statistics that are computed on the data. Let $f$ be any function that maps datasets to $\mathbb{R}^d$. The amount of noise depends on the *sensitivity* of $f$.

**Definition 2** (Sensitivity). *The* sensitivity *of a function $f$ is $\Delta_f = \max_{X,X' \in nbrs(X)} \|f(X) - f(X')\|_1$.*

We drop the subscript $f$ when it is clear from context. Our approach achieves differential privacy through the application of the Laplace mechanism.

**Definition 3** (Laplace Mechanism; Dwork et al. [1]). *Given a function $f$ that maps data sets to $\mathbb{R}^m$, the Laplace mechanism outputs the random variable $\mathcal{L}(X) \sim \mathrm{Lap}\left(f(X), \Delta_f/\epsilon\right)$ from the Laplace distribution, which has density $\mathrm{Lap}(z; u, b) = (2b)^{-m} \exp\left(-\|z - u\|_1/b\right)$. This corresponds to adding zero-mean independent noise $u_i \sim \mathrm{Lap}(0, \Delta_f/\epsilon)$ to each component of $f(X)$.*

A final important property of differential privacy is *post-processing* [15]; if an algorithm $\mathcal{A}$ is $\epsilon$-differentially private, then any algorithm that takes as input only the output of $\mathcal{A}$, and does not use the original data set $X$, is also $\epsilon$-differentially private.

## 3   Private Bayesian Inference in Exponential Families

We consider the canonical setting of Bayesian inference in an exponential family. The modeler posits a prior distribution $p(\theta)$, assumes the data $x_{1:n}$ is an iid sample from an exponential family model $p(x \mid \theta)$, and wishes to compute the posterior $p(\theta \mid x_{1:n})$. An exponential family in natural parameterization has density
$$p(x \mid \eta) = h(x) \exp\left(\eta^\top t(x) - A(\eta)\right),$$
where $\eta$ are the natural parameters, $t(x)$ is the sufficient statistic, $A(\eta) = \int h(x) \exp\left(\eta^\top t(x)\right) dx$ is the log-partition function, and $h(x)$ is the base measure. The density of the full data is
$$p(x_{1:n} \mid \eta) = h(x_{1:n}) \exp\left(\eta^\top t(x_{1:n}) - nA(\eta)\right),$$
where $h(x_{1:n}) = \prod_{i=1}^n h(x_i)$ and $t(x_{1:n}) = \sum_{i=1}^n t(x_i)$. Notice that once normalizing constants are dropped, this density is dependent on the data only directly through the sufficient statistics, $s = t(x_{1:n})$.

We will write exponential families more generally as $p(x \mid \theta)$ to indicate the case when the natural parameters $\eta = \eta(\theta)$ depend on a different parameter vector $\theta$.

Every exponential family distribution has a conjugate prior distribution $p(\theta; \lambda)$[24] with hyperparameters $\lambda$. A conjugate prior has the property that, if it is used as the prior, then the posterior belongs to the same family, i.e., $p(\theta \mid x_{1:n}; \lambda) = p(\theta; \lambda')$ for some $\lambda'$ that depends only on $\lambda$, $n$, and the sufficient statistics $s$. We write this function as $\lambda' = \text{Conjugate-Update}(\lambda, s, n)$; our methods are not tied to the specific choice of conjugate prior, only that the posterior parameters can be calculated in this form. See supplementary material for a general form of Conjugate-Update.

### 3.1   Release Algorithm: Noisy Sufficient Statistics

If privacy were not a concern, the Bayesian modeler would simply compute the sufficient statistics $s = t(x_{1:n})$ and use them to update the posterior beliefs. However, to maintain privacy, the modeler must access the sensitive data only through a randomized release mechanism $\mathcal{A}$. As a result, in order to obtain proper posterior beliefs the modeler must account for the randomization of the release mechanism by performing inference.

We take the simple approach of releasing noisy sufficient statistics via the Laplace mechanism, as in [12, 13, 19]. Sufficient statistics are a natural quantity to release. They are an "information bottleneck"—a finite-dimensional quantity that captures all the relevant information about $\theta$. The released value is $y = \mathcal{A}(x_{1:n}) \sim \mathrm{Lap}(s, \Delta_s/\epsilon)$. Because $s = t(x_{1:n}) = \sum_{i=1}^n t(x_i)$ is a sum over individuals, the sensitivity is $\Delta_s = \max_{x,x' \in \mathbb{R}^d} \|t(x) - t(x')\|_1$. When $t(\cdot)$ is unbounded this quantity becomes infinite; we will modify the release mechanism so the sensitivity is finite (Sec. 3.3).

### 3.2 Basic Inference Approach: Bounded Sufficient Statistics

The goal of the inference algorithm $\mathcal{P}$ is to compute $p(\theta \mid y)$. We first develop the basic approach for the simpler case when $t(x)$ is bounded, and then extend both $\mathcal{A}$ and $\mathcal{P}$ to handle the unbounded case. The full joint distribution of the probability model can be expressed as:

$$p(\theta, s, y) = p(\theta)\, p(s \mid \theta)\, p(y \mid s),$$

where $p(\theta) = p(\theta; \lambda)$ is a conjugate prior and the goal is to compute a representation of $p(\theta \mid y) \propto \int_s p(\theta, s, y) ds$ by integrating over the sufficient statistics.

We will develop a Gibbs sampler to sample from this distribution. There are two main challenges. First, the distribution $p(s \mid \theta)$ is obtained by marginalizing over the data sample $x_{1:n}$, and is usually not known in closed form. We will address this with an asymptotically correct normal approximation. Second, when resampling $s$ within the Gibbs algorithm, we require the full conditional distribution of $s$ given the other variables, which is proportional to $p(s|\theta)p(y \mid s)$. Care must be taken to make it easy to sample from this conditional distribution. We address this via variable augmentation. We discuss our approach to both challenges in detail below.

**Normal approximation of** $p(s \mid \theta)$**.**  The exact form of the sufficient statistic distribution $p(s \mid \theta)$ is obtained by marginalizing over the data:

$$p(s \mid \theta) = \int_{t^{-1}(s)} p(x_{1:n} \mid \theta) dx_{1:n}, \qquad t^{-1}(s) := \left\{ x_{1:n} : t(x_{1:n}) = s \right\}.$$

In general, the exact form of this distribution is not available. In some cases, it is—for example if $x \sim \text{Bernoulli}(\theta)$ then $s \sim \text{Binomial}(n, \theta)$—but even then it may not lead to a tractable full conditional for $s$.

Properties of exponential families pave the way toward a general approach that always leads to a tractable full conditional. By the central limit theorem (CLT), because $s = \sum_i t(x_i)$ is a sum of iid random variables, it is asymptotically normal. It can be approximated as $p(s \mid \theta) \approx \mathcal{N}(s; n\mu, n\Sigma)$, where $\mu = \mathbb{E}[t(x)]$ and $\Sigma = \text{Var}[t(x)]$ are the mean and variance of the sufficient statistic of a single individual. This approximation is asymptotically correct: $\frac{1}{\sqrt{n}}(s - n\mu) \xrightarrow{D} \mathcal{N}(0, \Sigma)$ [25]. The quantities $\mu$ and $\Sigma$ can be computed using well-known properties of exponential families [25]:

$$\mu = \mathbb{E}[t(x)] = \frac{\partial}{\partial \eta^\top} A(\eta), \qquad \Sigma = \text{Var}[t(x)] = \frac{\partial^2}{\partial \eta \partial \eta^\top} A(\eta), \tag{1}$$

where $\eta = \eta(\theta)$ is the natural parameter.

Note that we will *not* use this approximation for Gibbs updates of $\theta$. Instead, we will compute the conditional $p(\theta \mid s)$ using standard conjugacy formulas. In this sense, we maintain two views of the joint distribution $p(\theta, s)$—when updating $\theta$, it is the standard exponential family model, which leads to conjugate updates; when updating $s$, it is approximated as $p(\theta)\mathcal{N}(s; n\mu, s\Sigma)$, which will lead to simple updates when combined with a variable augmentation technique.

**Variable augmentation for** $p(y \mid s)$**.**  We seek a tractable form for the full conditional of $s$ under the normal approximation, which is the product of a normal density and a Laplace density:

$$p(s \mid \theta, y) \propto \mathcal{N}(s; n\mu, n\Sigma)\, \text{Lap}(y; s, \Delta_s/\epsilon).$$

A similar situation arises in the Bayesian Lasso [22], and we will employ the same variable augmentation trick. A Laplace random variable $z \sim \text{Lap}(u, b)$ can be written as a scale mixture of normals by introducing a latent variable $\sigma^2 \sim \text{Exp}(1/(2b^2))$, i.e., the distribution with density $1/(2b^2) \exp\left(-\sigma^2/(2b^2)\right)$ and letting $z \sim \mathcal{N}(u, \sigma^2)$. We apply this separately to each dimension of the vector $y$ so that:

$$\sigma_j^2 \sim \text{Exp}\left( \frac{\epsilon^2}{2\Delta_s^2} \right), \quad y \sim \mathcal{N}\big(s, \text{diag}(\sigma^2)\big).$$

| **Algorithm 1** Gibbs Sampler, Bounded $\Delta_s$ | **Subroutine** NormProduct |
|---|---|
| 1: Initialize $\theta, s, \sigma^2$ | 1: **input:** $\mu_1, \Sigma_1, \mu_2, \Sigma_2$ |
| 2: **repeat** | 2: $\Sigma_3 = \left(\Sigma_1^{-1} + \Sigma_2^{-1}\right)^{-1}$ |
| 3:      $\theta \sim p(\theta; \lambda')$ where $\lambda' = \text{Conjugate-Update}(\lambda, s, n)$ | 3: $\mu_3 = \Sigma_3 \left(\Sigma_1^{-1}\mu_1 + \Sigma_2^{-1}\mu_2\right)$ |
| 4:      Calculate $\mu = \mathbb{E}[s]$ and $\Sigma = \text{Var}[s]$ (e.g., use Eq. (1)) | 4: **return:** $\mathcal{N}(\mu_3, \Sigma_3)$ |
| 5:      $s \sim \text{NormProduct}\left(n\mu, n\Sigma, y, \text{diag}(\sigma^2)\right)$ | |
| 6:      $1/\sigma_j^2 \sim \text{InverseGaussian}\left(\frac{\epsilon}{\Delta_s |y-s|}, \frac{\epsilon^2}{\Delta_s^2}\right)$ | |

**The Gibbs Sampler.** After the normal approximation and variable augmentation, the generative process is as shown to the right. The final Gibbs sampling algorithm is shown in Algorithm 1. Note that the update for $\theta$ is based on conjugacy in the exact distribution $p(\theta, s)$, while the update for $s$ uses the density of the generative process to the right, so that $p(s \mid \theta, \sigma^2, y) \propto p(s \mid \theta)\, p(y \mid \sigma^2, s)$, which is a product of two normal densities

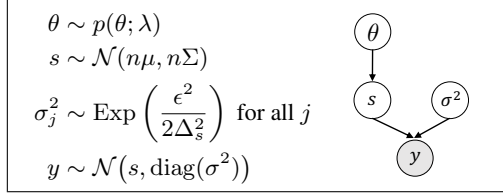

$$\mathcal{N}(s; n\mu, n\Sigma)\,\mathcal{N}\left(y; s, \text{diag}(\sigma^2)\right) \propto \mathcal{N}(s; \mu_s, \Sigma_s),$$

where $\mu_s$ and $\Sigma_s$ are are defined in Algorithm 1 [26]. The update for $\sigma^2$ follows Park and Casella [22]; the inverse Gaussian density is $\text{InverseGaussian}(x; m, v) = \sqrt{v/(2\pi x^3)} \exp\left(-v(x-m)^2/(2m^2 x)\right)$. Full derivations are given in the supplement.

### 3.3   Unbounded Sufficient Statistics and Truncated Exponential Families

The Laplace mechanism does not apply when the sufficient statistics are unbounded, because $\Delta_s = \max_{x,y} \|t(x) - t(y)\|_1 = \infty$. Thus, we need a new release mechanism $\mathcal{A}$ and inference algorithm $\mathcal{P}$. We present a solution for the case when $x$ is univariate. All elements of the solution can generalize to higher dimensions, except that one step will have running time that is exponential in $d$; we leave improvement of this to future work and focus on the simpler univariate case.

**Release mechanism.** Our solution is to truncate the support of the (now univariate) $p(x \mid \theta)$ to $x \in [a, b]$, where $a$ and $b$ are finite bounds provided by the modeler. If the modeler cannot select bounds *a priori*, they may be selected privately as a preliminary step using a variant of the exponential mechanism (see `PrivateQuantile` in Smith [27]).[3] Then, given truncation bounds, the data owner redacts individuals where $x_i \notin [a, b]$ and reports the truncated sufficient statistics $\hat{s} = \sum_{i=1}^n \mathbf{1}_{[a,b]}(x_i) \cdot t(x_i)$ where $\mathbf{1}_S(x)$ is the indicator function of the set $S$. The sensitivity of $\hat{s}$ is now $\Delta_{\hat{s}} = \max_{x,y \in \mathbb{R}} \|\hat{t}(x) - \hat{t}(y)\|_1$ where $\hat{t}(x) = \mathbf{1}_{[a,b]}(x)\, t(x)$. An easy upper bound for this quantity (see supplement) is:

$$\Delta_{\hat{s}} \leq \sum_{j=1}^d \max\left\{ \max_{x \in [a,b]} \big|t_j(x)\big|, \max_{x,y \in [a,b]} \big|t_j(x) - t_j(y)\big| \right\},$$

where $t_j(x)$ is the $j$th component of the sufficient statistics. The bounds $[a, b]$ will be selected so this quantity is bounded. The released value is $y \sim \text{Lap}(\hat{s}, \Delta_{\hat{s}}/\epsilon)$.

**Inference: Truncated Exponential Family.** Several new challenges arise for inference. The quantity $\hat{s}$ is no longer a sufficient statistic for the model $p(x \mid \theta)$, and we will need new insights to understand $p(\hat{s} \mid \theta)$ and $p(\theta \mid \hat{s})$. Since $\hat{s}$ is a sum over individuals where $x_i \in [a, b]$, it will be useful to examine the probability of the event $x \in [a, b]$ as well as the conditional distribution of $x$ given this event. To facilitate a general development, assume a generic truncation interval $[v, w]$, not necessarily

equal to $[a, b]$. Let $F(x; \theta) = \int_{-\infty}^{x} p(x \mid \theta) dx$ be the CDF of the original (univariate) exponential family model. It is clear that $\Pr(x \in [v, w]) = F(w; \theta) - F(v; \theta)$. The conditional distribution of $x$ given $x \in [v, w]$ is a *truncated* exponential family, which, in its natural parameterization is:

$$\hat{p}(x \mid \eta) = \mathbf{1}_{[v,w]}(x) \, h(x) \, \exp\left(\eta^T t(x) - \hat{A}(\eta)\right), \quad \hat{A} = \int_{v}^{w} h(x) \exp\left(\eta^T t(x)\right) dx. \quad (2)$$

Note that this is still an exponential family model (with a modified base measure), and all of the standard results apply, such as the existence of a conjugate prior and the formulas in Eq. (1) for the mean and variance of $t(x)$ under the truncated distribution.

**Random sum CLT for $p(\hat{s} \mid \theta)$.** We would like to again apply an asymptotic normal approximation for $\hat{s}$, but we do not know how many individuals fall within the truncation bounds. The "random sum CLT" of Robbins [28] applies to the setting where the number of terms in the sum is itself a random variable. The sum can be rewritten as $\hat{s} = \sum_{k=1}^{N} t(x_{i_k})$, where $\{i_1, \ldots, i_N\}$ is the set of indices of individuals with data inside the truncation bounds, i.e., the indices such that $x_{i_k} \in [v, w]$. The number $N$ is now a random variable distributed as $N \sim \text{Binom}(n, q)$, where $q = F(w; \theta) - F(v; \theta)$.

**Proposition 1.** *Let $\hat{\mu} = \mathbb{E}_{\hat{p}}[t(x)]$ and $\hat{\Sigma} = \text{Var}_{\hat{p}}[t(x)]$ be the mean and variance of $t(x)$ in the truncated exponential family. Then $\hat{s} = \sum_{k=1}^{N} t(x_{i_k})$ is asymptotically normal with mean and variance:*

$$\mathbf{m} := \mathbb{E}[\hat{s}] = \mathbb{E}[N]\hat{\mu} = nq\hat{\mu},$$

$$\mathbf{V} := \text{Var}(\hat{s}) = \mathbb{E}[N]\hat{\Sigma} + \text{Var}[N]\hat{\mu}\hat{\mu}^\top = nq\hat{\Sigma} + nq(1-q)\hat{\mu}\hat{\mu}^\top.$$

*Specifically, $\frac{1}{\sqrt{n}}(\hat{s} - \mathbf{m}) \xrightarrow{D} \mathcal{N}(0, \bar{\Sigma})$ as $n \to \infty$, where $\bar{\Sigma} = \mathbf{V}/n = q\hat{\Sigma} + q(1-q)\hat{\mu}\hat{\mu}^\top$.*

*Proof.* Each term of the sum has mean $\hat{\mu}$ and variance $\hat{\Sigma}$, and the number of terms is $N \sim \text{Binom}(n, q)$. The result follows from Robbins [28]. $\square$

**Computing $\hat{\mu}$ and $\hat{\Sigma}$ by automatic differentiation (autodiff).** To use the normal approximation we need to compute $\hat{\mu}$ and $\hat{\Sigma}$.

**Lemma 1.** *Let $p(x \mid \theta)$ be a univariate exponential family model and let $\hat{p}(x \mid \theta)$ be the corresponding exponential family model truncated to generic interval $[v, w]$. Then*

$$\hat{\mu} = \mathbb{E}_{\hat{p}}[t(x)] = \mathbb{E}_p[t(x)] + \frac{\partial}{\partial \eta^T} \log\left(F(w; \eta) - F(v; \eta)\right) \quad (3)$$

$$\hat{\Sigma} = \text{Var}_{\hat{p}}[t(x)] = \text{Var}_p[t(x)] + \frac{\partial^2}{\partial \eta \partial \eta^T} \log\left(F(w; \eta) - F(v; \eta)\right) \quad (4)$$

*Proof.* It is straightforward to derive from Eq. (2) that $\hat{A}(\eta) = A(\eta) + \log\left(F(w; \eta) - F(v; \eta)\right)$. The result follows from applying Eq. (1) to this expression for $\hat{A}(\eta)$. See the supplement for derivation of $\hat{A}(\eta)$ and proof of this lemma. $\square$

We will use Equations (3) and (4) to compute $\hat{\mu}$ and $\hat{\Sigma}$ by using autodiff to compute the desired derivatives. If the mean and variance $\mathbb{E}_p[t(x)]$ and $\text{Var}_p[t(x)]$ of the untruncated distribution are not known, we can apply autodiff to compute them as well using Eq. (1).

When $x$ is multivariate, analogous expressions can be derived for $\hat{\mu}$ and $\hat{\Sigma}$. The adjustment factors will include multivariate CDFs, with a number of terms that grow exponentially in $d$. This is currently the main limitation in applying our methods to multivariate models with unbounded sufficient statistics.

**Conjugate updates for $p(\theta \mid \hat{s})$.** The final issue is the distribution $p(\theta \mid \hat{s})$, which is no longer characterized by conjugacy because $\hat{s}$ are not the full sufficient statistics. We again turn to variable augmentation. Let $\hat{s}_\ell = \sum_{i=1}^{n} \mathbf{1}_{[-\infty, a]} t(x_i)$ and $\hat{s}_u = \sum_{i=1}^{n} \mathbf{1}_{[b, \infty]} t(x_i)$ be the sufficient statistics for the individuals that fall in the lower portion $[-\infty, a]$ and upper portion $[b, \infty]$ of the support of $x$, respectively. We will instantiate $\hat{s}_\ell$ and $\hat{s}_u$ as latent variables and model their distributions using the

| **Algorithm 2** Gibbs Sampler, Unbounded $\Delta_s$ | **Algorithm 3** RS-CLT |
|---|---|
| 1: Initialize $\theta, \hat{s}, \sigma^2, a, b$ | 1: **input:** $\theta, v, w$ |
| 2: $[v_\ell, w_\ell] \leftarrow [-\infty, a]$ | 2: $q \leftarrow F(b; w) - F(a; v)$ |
| 3: $[v_c, w_c] \leftarrow [a, b]$ | 3: $\hat{\mu}, \hat{\Sigma} \leftarrow$ autodiff of Eqns. 3, 4 |
| 4: $[v_u, w_u] \leftarrow [b, \infty]$ | 4: $\mathbf{m} \leftarrow nq$ |
| 5: **repeat** | 5: $\mathbf{V} \leftarrow nq\hat{\Sigma} + nq(1-q)\hat{\mu}\hat{\mu}^\top$ |
| 6: $\quad \mathbf{m}_r, \mathbf{V}_r \leftarrow$ RS-CLT$(\theta, v_r, w_r)$ for $r \in \{\ell, c, u\}$ | 6: **return: m, V** |
| 7: $\quad \mathbf{m}'_c, \mathbf{V}'_c \leftarrow$ NormProduct $\left(\mathbf{m}_c, \mathbf{V}_c, y, \text{diag}\left(\sigma^2\right)\right)$ | |
| 8: $\quad s \sim \mathcal{N}(\mathbf{m}_\ell + \mathbf{m}'_c + \mathbf{m}_u, \mathbf{V}_\ell + \mathbf{V}'_c + \mathbf{V}_u)$ | |
| 9: $\quad \theta \sim p(\theta; \lambda')$ where $\lambda' = $ Conjugate-Update$(\lambda, s, n)$ | |
| 10: $\quad$ Recalculate $\mathbf{m}_c$ and $\mathbf{V}_c$, then draw $\hat{s}_c \sim \mathcal{N}(\mathbf{m}_c, \mathbf{V}_c)$ | |
| 11: $\quad 1/\sigma_j^2 \sim$ InverseGaussian$\left(\frac{\epsilon}{\Delta_{\hat{s}}|y - \hat{s}_c|}, \frac{\epsilon^2}{\Delta_{\hat{s}}^2}\right)$ | |
| 12: **until** | |

random sum CLT approximation from Prop. 1 and Lemma 1 (but with different truncation bounds). Let $\hat{s}_c = \hat{s}$ be the sufficient statistics for the "center" portion, and define the three truncation intervals as $[v_\ell, w_\ell] = [-\infty, a]$, $[v_c, w_c] = [a, b]$ and $[v_u, w_u] = [b, \infty]$. The full sufficient statistics are equal to $s = \hat{s}_\ell + \hat{s}_c + \hat{s}_u$. Conditioned on all other variables, *each* component is multivariate normal, so the sum $s$ is also multivariate normal. We can therefore sample $s$ and then sample from $p(\theta \mid s)$ using conjugacy. We will also need to draw $\hat{s}_c$ separately to be used to update $\sigma^2$.

**The Gibbs Sampler.** The (approximate) generative process in the unbounded case is:

$$\theta \sim p(\theta; \lambda),$$
$$\hat{s}_r \sim \mathcal{N}\big(\mathbf{m}_r, \ \mathbf{V}_r\big), \text{ for } r \in \{\ell, c, u\} \text{ where } \mathbf{m}_r, \mathbf{V}_r = \text{RS-CLT}(\theta, v_r, w_r)$$
$$\sigma_j^2 \sim \text{Exp}\left(\frac{\epsilon^2}{2\Delta_{\hat{s}}^2}\right) \text{ for all } j \ ,$$
$$y \sim \mathcal{N}\big(\hat{s}_c, \text{diag}(\sigma^2)\big).$$

The Gibbs sampler to sample from this distribution is given in Algorithm 2. Note that in Line 8 we employ rejection sampling in which sufficient statistics are sampled until the values drawn are valid for the given data model, e.g., $s$ must be positive for the binomial distribution. The RS-CLT algorithm to compute parameters of the random sum CLT is shown in Algorithm 3.

## 4 Experiments

We design experiments to measure the *calibration* and *utility* of our method for posterior inference. We conduct experiments for the binomial model with beta prior, the multinomial model with Dirichlet prior, and the exponential model with gamma prior. The last model is unbounded and requires truncation; we set the bounds to keep the middle 95% of individuals, which is reasonable to assume known a priori for some cases, such as modeling human height.

**Methods.** We run our Gibbs sampler for 5000 iterations after 2000 burnin iterations (see supplementary material for convergence results), which we compare to two baselines. The first method uses the same release mechanism as our Gibbs sampler and performs conjugate updates using the noisy sufficient statistics [12, 13]. This method converges to the true posterior as $n \to \infty$ because the Laplace noise will eventually become negligible compared to sampling variability [12]. However, the noise is not negligible for moderate $n$; we refer to this method as "naive". For truncated models we allow the naive method to "cheat" by accessing the noisy *untruncated* sufficient statistics $s$. Thus the method is not private, and receives strictly more information than our Gibbs sampler, but with the same magnitude noise. This allows us to demonstrate miscalibration without highly technical modifications to the baseline method to be able to deal with truncated sufficient statistics.

The second baseline is a version of the one-posterior sampling (OPS) mechanism [11–13], which employs the exponential mechanism [29] to release samples from a privatized posterior. We release 100 samples using the method of [12], each with $\epsilon_{ops} = \epsilon/100$, such that the entire algorithm achieves $\epsilon$-differential privacy. Private MCMC sampling [11] is a more sophisticated method to release multiple samples from a privatized posterior and could potentially make better use of the privacy budget; however, private MCMC will also necessarily be miscalibrated, and only achieves the weaker privacy guarantee of $(\epsilon, \delta)$-differential privacy for $\delta > 0$, so would not be direct comparable to our method. OPS serves as a suitable baseline that achieves $\epsilon$-differential privacy. We include OPS only for experiments on the binomial model, for which it requires the support of $\theta$ to be truncated to $[a_0, 1 - a_0]$ where $a_0 > 0$. We set $a_0 = 0.1$.

We also include a non-private posterior for comparison, which performs conjugate updates using the non-noisy sufficient statistics.

**Evaluation.** We evaluate both the *calibration* and *utility* of the posterior. For calibration we adapt a method of Cook et al. [30]: the idea is to draw iid samples $(\theta_i, x_i)$ from the joint model $p(\theta)p(x \mid \theta)$, and conduct posterior inference in each trial. Let $F_i(\theta)$ be the CDF of the *true* posterior $p(\theta \mid x_i)$ in trial $i$. Then we know that $U_i = F_i(\theta_i)$ is uniformly distributed, because $\theta_i \sim p(\theta \mid x_i)$ (see supplementary material). In other words, the actual parameter $\theta_i$ is equally likely to land at any quantile of the posterior. To test the posterior inference procedure, we instead compute $U_i$ as the quantile at which $\theta_i$ lands within a set of samples from the *approximate* posterior. After $M$ trials of the whole procedure we test for uniformity of $U_{1:M}$ using the Kolmogorov-Smirnov goodness-of-fit test [31], which measures the maximum distance between the empirical CDF of $U_{1:M}$ and the uniform CDF; lower values are better and zero corresponds to perfect uniformity. We also visualize the empirical CDFs to assess calibration qualitatively.

Higher utility of a private posterior is indicated by closeness to the non-private posterior, which we measure with *maximum mean discrepancy* (MMD), a kernel-based statistical test to determine if two sets of samples are drawn from different distributions [32]. Given $m$ i.i.d. samples $(p, q) \sim P \times Q$, an unbiased estimate of the MMD is

$$\text{MMD}^2(P, Q) = \frac{1}{m(m-1)} \sum\nolimits_{i \neq j}^{m} \left( k(p_i, p_j) + k(q_i, q_j) - k(p_i, q_j) - k(p_j, q_i) \right),$$

where $k$ is a continuous kernel function; we use a standard normal kernel. The higher the value the more likely the two samples are drawn from different distributions.

**Results.** Figure 1a shows the results for three models and varying $n$ and $\epsilon$. Our method (Gibbs) achieves the same calibration level as non-private posterior inference for all settings. The naive method ignores noise and is too confident about parameter values implied by treating the noisy sufficient statistics as true ones; it is only well-calibrated with increasing $n$ and $\epsilon$ when noise becomes negligible relative to population size. OPS is not calibrated because it samples from an over-dispersed version of $p(\theta \mid x)$.

Figure 1b shows the empirical CDF plots for $n = 1000$ and $\epsilon = 0.01$. Our method and the non-private method are both perfectly calibrated. The naive method's over-confidence in the wrong sufficient statistics causes its posterior to usually be too tight at the wrong value; thus the true parameter always lies in a tail of the approximate posterior, so too much mass is placed near 0 and 1. OPS shows the opposite behavior: its posterior is always too diffuse, so the true parameter lies close to the middle. For multinomial we show measures only for the parameter of the first category, but results hold for all categories.

Figure 1c shows the MMD test statistic between each method and the non-private posterior, used as a measure of utility. Our method consistently achieves utility at least as good as the naive method for binomial and multinomial models. We omit OPS, which is never calibrated. For the exponential model (not shown) we did not obtain conclusive utility comparisons due to the lack of a naive baseline that properly handles truncation; the "cheating" naive method from our calibration experiments sometimes attains higher utility than our method, and sometimes lower, but this comparison is not meaningful because it receives strictly more information.

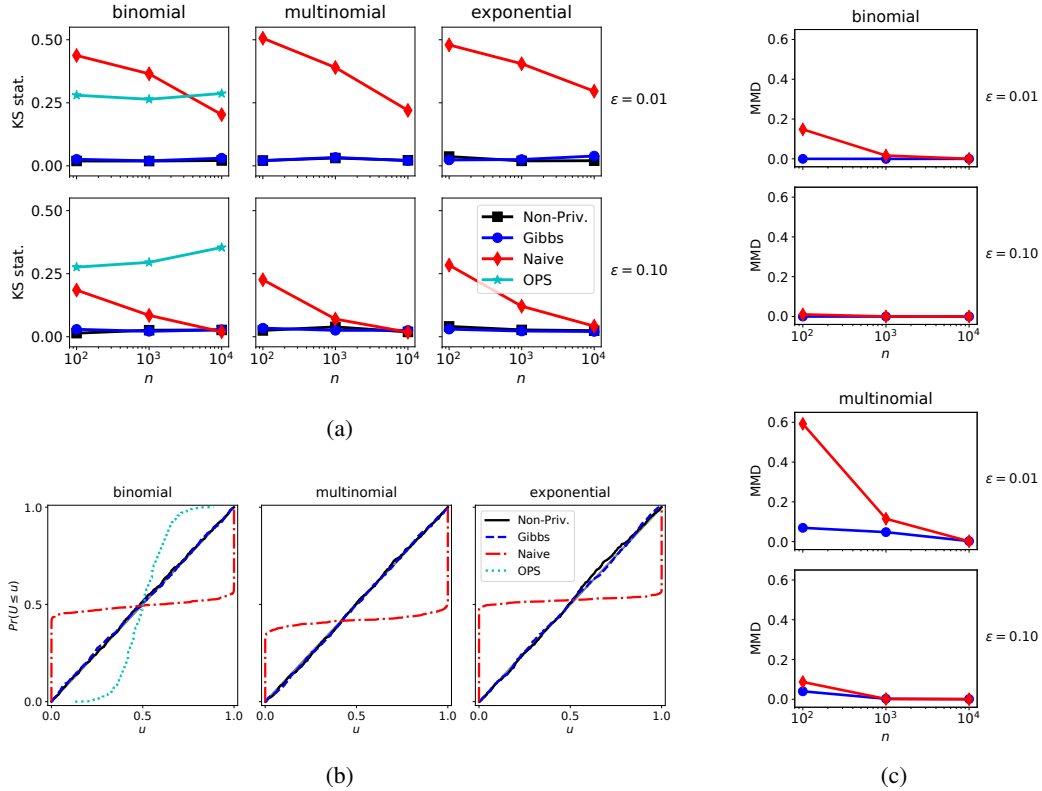

Figure 1: (a) Calibration as Kolmogorov-Smirnov statistic vs. number of individuals at $\epsilon = [0.01, 0.10]$ for binomial, multinomial, and exponential models. (b) Empirical CDF plots at $(n = 1000; \epsilon = 0.01)$ for binomial, multinomial, and exponential models. (c) Utility as MMD with non-private posterior vs. number of individuals at $\epsilon = [0.01, 0.10]$ for binomial and multinomial models.

## 5 Conclusion

We presented a Gibbs sampling approach for private posterior inference in exponential family models. Rather than trying to approximate the posterior of $p(\theta \mid x_{1:n})$, we divide our procedure into a private *release mechanism* $y = \mathcal{A}(x_{1:n})$ and an *inference algorithm* $\mathcal{P}$ that computes $p(\theta \mid y)$. The release mechanism is designed to facilitate inference. We develop a general-purpose Gibbs sampler that applies to any exponential family model that has bounded sufficient statistics; a truncated version applies to univariate models with unbounded sufficient statistics. The Gibbs sampler uses general properties of exponential families to approximate the distribution of the sufficient statistics, and therefore avoids the need to reason about individuals. Promising lines of future work are to develop efficient methods for multivariate exponential families with unbounded sufficient statistics, and to develop methods for conditional models based on exponential families, such as generalized linear models.

### Acknowledgments

This material is based upon work supported by the National Science Foundation under Grant Nos. 1522054 and 1617533.

## Footnotes

[1]There are remaining technical challenges for multivariate models with unbounded sufficient statistics that we leave for future work.

[2]This variant assumes $n$ remains fixed, which is sometimes called *bounded* differential privacy [23].

[3]Selecting truncation bounds will consume some of the privacy budget and modify the release mechanism $\mathcal{A}$. We do not consider inference with respect to this part of the release mechanism.

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
