[Supplementary Material · expfam_supplementary.pdf]

# A Properties of Exponential Families

## A.1 Form of Conjugate-Update$(\lambda, x_{1:n})$

Following Diaconis and Ylvisaker [24], the prior is

$$p(\eta \mid \lambda) = h(\lambda) \exp\left(\lambda_1^T \eta - \lambda_2 A(\eta) - B(\lambda)\right),$$

where the parameters are $\lambda = [\lambda_1, \lambda_2]$ and sufficient statistics are $[\eta, -A(\eta)]$

The posterior after observing $x_{1:n}$ is

$$p(\eta \mid \lambda, x_{1:n}) = h(\lambda') \exp\left(\lambda_1'^T x - \lambda_2' A(\eta) - B(\lambda')\right)$$

$$\lambda_1' = \lambda_1 + \sum_i t(x_i)$$

$$\lambda_2' = \lambda_2 + n$$

Define above updates as $\lambda' = \text{Conjugate-Update}(\lambda, x_{1:n})$

### A.1.1 Proof of Log-Partition Function of Truncated Distribution used in Lemma 1

**Claim**:

$$\hat{A}(\eta) = A(\eta) + \log\left(F(w; \eta) - F(v; \eta)\right)$$

**Proof**:

$$\exp\left(\hat{A}(\eta)\right) = \int_v^w h(x) \exp\left(\eta^T t(x)\right) dx$$

$$= \exp\left(A(\eta)\right) \int_v^w h(x) \exp\left(\eta^T t(x) - A(\eta)\right) dx$$

$$= \exp\left(A(\eta)\right)\left(F(w; \theta) - F(v; \theta)\right)$$

### A.1.2 Proof of Lemma 1: Mean and Variance of $t(x)$ in truncated distribution

**Claim**

$$\mathbb{E}_{\hat{p}}[t(x)] = \mathbb{E}_p[t(x)] + \frac{\partial}{\partial \eta^T} \log\left(F(w; \eta) - F(v; \eta)\right)$$

$$\text{Var}_{\hat{p}}[t(x)] = \text{Var}_p[t(x)] + \frac{\partial^2}{\partial \eta \partial \eta^T} \log\left(F(w; \eta) - F(v; \eta)\right)$$

**Proof**:

$$\mathbb{E}_{\hat{p}}[t(x)] = \frac{\partial}{\partial \eta^T} \hat{A}(\eta)$$

$$= \frac{\partial}{\partial \eta^T}\left(A(\eta) + \log\left(F(w; \eta) - F(v; \eta)\right)\right)$$

$$= \frac{\partial}{\partial \eta^T} A(\eta) + \frac{\partial}{\partial \eta^T} \log\left(F(w; \eta) - F(v; \eta)\right)$$

$$= \mathbb{E}_p[t(x)] + \frac{\partial}{\partial \eta^T} \log\left(F(w; \eta) - F(v; \eta)\right)$$

The proof for $\text{Var}_{\hat{p}}[t(x)]$ is similar.

# B Derivation of $\sigma^2$ Gibbs update

We fully derive the Gibbs update for the noise variance $\sigma^2$ of the augmented model as stated in Park and Casella [22]. We represent the Laplace distribution with scale $b = \Delta_s/\epsilon$ as a scale mixture of normals, i.e. a zero-mean normal with an exponential prior on the variance:

$$p(z \mid b) = \frac{1}{2b} \exp\left(-\frac{|z|}{b}\right) = \int_0^\infty \underbrace{\frac{1}{\sqrt{2\pi\sigma^2}} \exp\left(-\frac{z^2}{2\sigma^2}\right)}_{p(z|\sigma^2)} \cdot \underbrace{\ell \exp\left(-\ell\sigma^2\right)}_{p(\sigma^2|b)} d\sigma^2, \quad \ell = 1/2b^2$$

For clarity we have written the exponential rate as $\ell = 1/2b^2$. Also recall that the noise $z$ corresponds to the difference $y - s$ between the noisy and non-noisy sufficient statistics in our model. As per Park and Casella [22] we can write the conditional update for $\sigma^2$ as a Wald distribution (inverse-Gaussian) with the change of variable $t = 1/\sigma^2$:

$$\begin{aligned}
p_t\left(t \mid z, \ell\right) &= \left|\frac{d}{dt}\frac{1}{t}\right| \cdot p_{\sigma^2}\left(\frac{1}{t} \mid z, \ell\right) \\
&= \frac{1}{t^2} \cdot p_{\sigma^2}\left(\frac{1}{t} \mid z, \ell\right) \\
&= \frac{1}{t^2} \cdot \frac{1}{\sqrt{2\pi\frac{1}{t}}} \exp\left(-\frac{z^2}{2\frac{1}{t}}\right) \cdot \ell \exp\left(-\frac{\ell}{t}\right) \\
&\propto \frac{1}{\sqrt{t^3}} \exp\left(-\frac{z^2}{2}t - \frac{\ell}{t}\right)
\end{aligned}$$

`numpy.random.Wald` is a two-parameter (mean and scale) implementation of inverse-Gaussian. Its pdf is

$$\begin{aligned}
\text{Wald}(t; \mu, \gamma) &= \frac{\gamma}{\sqrt{2\pi t^3}} \exp\left(-\frac{\gamma(t-\mu)^2}{2\mu^2 t}\right) \\
&\propto \frac{1}{\sqrt{t^3}} \exp\left(-\frac{\gamma(t-\mu)^2}{2\mu^2 t}\right) \\
&= \frac{1}{\sqrt{t^3}} \exp\left(-\frac{\gamma t^2 - 2\gamma\mu t + \gamma\mu^2}{2\mu^2 t}\right) \\
&= \frac{1}{\sqrt{t^3}} \exp\left(-\frac{\gamma}{2\mu^2}t + \frac{\gamma}{\mu} - \frac{\gamma}{2t}\right) \\
&\propto \frac{1}{\sqrt{t^3}} \exp\left(-\frac{\gamma}{2\mu^2}t - \frac{\gamma}{2t}\right)
\end{aligned}$$

Then matching parameters we have

$$\begin{aligned}
\gamma &= 2\ell \\
&= \frac{1}{b^2}
\end{aligned}$$

and

$$\begin{aligned}
\frac{\gamma}{\mu^2} &= z^2 \\
\mu &= \sqrt{\frac{\gamma}{z^2}} = \frac{1}{bz}
\end{aligned}$$

So we draw $t$ from

$$p\left(t \mid z, b\right) = \text{Wald}\left(t; \frac{1}{bz}, \frac{1}{b^2}\right)$$

and set $\sigma^2 = 1/t$.

## C  Sensitivity of Sufficient Statistics in Truncated Model

Recall that $\hat{t}(x) = \mathbf{1}_{[v,w]}(x)\,t(x)$. Then

$$
\begin{aligned}
\Delta_{\hat{s}} &= \max_{x,y\in\mathbb{R}} \|\hat{t}(x) - \hat{t}(y)\|_1 \\
&= \max_{x,y\in\mathbb{R}} \sum_j |\hat{t}_j(x) - \hat{t}_j(y)| \\
&\leq \sum_j \max_{x,y\in\mathbb{R}} |\hat{t}_j(x) - \hat{t}_j(y)| \\
&= \sum_j \max\left\{ \max_{x\in[v,w],y\notin[v,w]} |\hat{t}_j(x) - \hat{t}_j(y)|,\ \max_{x,y\in[v,w]} |\hat{t}_j(x) - \hat{t}_j(y)| \right\} \\
&= \sum_j \max\left\{ \max_{x\in[v,w]} |t_j(x)|,\ \max_{x,y\in[v,w]} |t_j(x) - t_j(y)| \right\}
\end{aligned}
$$

## D  Proof of uniformity of CDF transform used by Cook et al. [30]

**Claim**: Let $X$ be a random variable with CDF $F$. The random variable $U = F(X)$ is uniformly distributed.

**Proof**:

$$
\begin{aligned}
\Pr(U \leq u) &= \Pr(F(X) \leq u) \\
&= \Pr\left(F^{-1}(F(X)) \leq F^{-1}(u)\right) \\
&= \Pr(X \leq F^{-1}(u)) \\
&= F\left(F^{-1}(u)\right) \\
&= u
\end{aligned}
$$

## E  Convergence of Gibbs Sampler

Figure 3 shows the progress of sampled model parameters over the course of 500 iterations for both binomial and exponential models. For both models the samples quickly converge to the vicinity of the true parameter.

Figure 3: Progress of Gibbs sampler parameters over iterations at $(n = 1000;\ \epsilon = 0.1)$ for binomial and exponential models.