[Reviews · NeurIPS 2018]

Reviewer 1



This paper proposes an approach for differentially private estimation of the posterior distribution in conjugate exponential-family models. Similar to previous "naive" approaches, it enforces privacy by adding Laplace-distributed noise to the sufficient statistic. Where a naive approach would treat this noisy statistic as true, the main contribution of this paper is a Gibbs sampling algorithm to integrate over uncertainty in the true statistic given the observed noisy statistic. This is the proper Bayesian procedure, and the experiments show that this yields better-calibrated posterior estimates than naive updating or one-posterior sampling (OPS). The paper is very clear, cleanly written and easy to follow; I found no obvious mistakes. It fills in a valuable part of the story for differentially private Bayesian inference. The proposed approach is sensible, appears to work well, and involves technically nontrivial contributions (CLT and variable-augmentation tricks to derive a tractable approximate Gibbs sampler, and handling unbounded sufficient statistics through truncation via autodiff and a random-sum CLT). This is a solid paper, certainly above the NIPS bar. My main concerns for this paper involve its impact and applicability. Differential privacy is a strong condition, and my understanding (as an outsider to the field) is that it's been difficult to practically apply differentially private algorithms with meaningful privacy guarantees. From a motivation standpoint, it would be nice to see some discussion of applications where differentially private Bayesian inference on small datasets might be useful. In the experiments, I would prefer to see some analysis of the utility of inference results, which the calibration experiments say nothing about. If I'm understanding the setup correctly, ignoring the data entirely and just returning the prior would be perfectly 'calibrated', but that is not a useful inference method. Are the posteriors actually reasonable for potential applications? You could quantify this through divergence metrics (e.g. KL) from the non-private posterior, mean posterior log-density of the true parameter (essentially the same thing as KL in this setting up to a constant), or even through storytelling -- what's a plausible application and setting of parameters where this method seems to give a useful posterior?

Reviewer 2



The work proposed Gibbs sampling algorithms for differential private Bayesian inference for general conjugate exponential families. The Laplace mechanism is used to release the sufficient statistics, and the paper aims to infer the posteriors given the randomly perturbated sufficient statistics. To this end, the work developed Gibbs sampling algorithm which uses Gaussian to approximate the conditional distribution of the statistics and variable augmentation for Laplace prior. To deal with unbounded statistics, the works truncated the statistics and adjust the Gaussian approximation to the (truncated) sufficient statistics to fulfill the Gibbs sampling. The experiment shows better posterior inference accuracy than competing private inference methods. This is an interesting cross-discipline work. Though the proposed model is very simple --- just introducing a release variable and injecting a Laplace prior, the inference is nontrival and the work combines a few interesting tricks (Gaussians & variable augmentation) to implement the Gibbs sampling. The results show promising results in terms of inference quality. I would like to know more about the experimental settings --- how many samples did you draw for each model? My concern is here: since your algorithm is not accurate Gibbs --- you use Gaussian to approximate the conditional distribution of the sufficient statistics, I am worrying that when the number of data points is too small, this approximation is bad and could affect the convergence. When you use truncated statistics become more severe. Is it possible to use Metropolis Hasting directly? If not, what will be the reason? Also, can you add more results showing the convergence behavior of the Gibbs and truncated Gibbs? The work is claimed to be “the first method for private Bayesian inference in exponential families that properly accounts for noise …”, but there seems other private Bayesian inference methods as well. However, I do not see a discuss of those works until the experimental section. It is better to add a “related work” selection to discuss and summarize status quo in private machine learning and Bayesian inference. Minor issue: algorithm complexity analysis is absent.

Reviewer 3



After considering author responses & other reviews, I keep my score unchanged. Some additional comments to the author responses On truncation bounds (rows 23-26): In my opinion, this definitely deserves to be mentioned in the paper. I agree that there are plenty of problems where a reasonable, say, 95%-bound is known accurately enough a priori. However, this is still an exception and more generally, especially in the strict privacy regime you are currently using, I would guess the bounds estimated under DP could actually be quite a bit off from the true values. Given that the truncation bound is a key hyperparamater that could completely break the method, I encourage you to add some (even simple) testing on this to the paper. Ops vs Private MCMC (44-47): I don't doubt that the calibration would be more or less off, the question in my mind is how much. But granted, I agree it's not generally feasible, or necessary in this case, to start comparing against (eps,delta)-methods as well. Other comments (14-17,41-43): Thank you, I'm happy with these. Differentially Private Bayesian Inference for Exponential Families The paper considers Bayesian posterior estimation for exponential family models under differential privacy (DP). More concretely, the main problem is how to infer a well-calibrated posterior distribution from noisy sufficient statistics, where the noise guarantees privacy by standard arguments. As a solution, the authors propose 2 versions of a Gibbs sampler: one for sampling from a general exponential family model with bounded sufficient statistics, and a variant that works with unbounded sufficient statistics but assumes the data is very low-dimensional. The main novelty in the approach is to include the DP noise properly into the modelling instead of simply "forgetting" it and treating the actually noisy posterior as a noiseless one, as is often done in existing literature. Intuitively, this should lead to better calibrated posteriors, which also is the case in the empirical tests shown in the paper. The idea is not totally unheard of (see e.g. references in the paper), but the formulation is nice, and the solution is novel. The problem as such is certainly an important one that needs to be solved properly, and the paper is a good step in the right direction. The paper is clearly written and the main ideas are easy to understand. The mathematical proofs provided in the paper are mostly already known or small modifications to existing proofs. There doesn't seem to be any major problems with the theory, and it seems well-suited to the problem. Pros: * Including the DP noise into the modelling is a very relevant idea, and there does not seem to be much papers on the topic. * The proposed method is novel, and seems to give well-calibrated posteriors with limited sample sizes under strict privacy. Cons: * The proposed solution has an exponential dependency on the dimensionality of the (unbounded) sufficient statistics (as noted by the authors). * No theory or testing on how the assumed bounds [a,b] for s_c affect the results. * Fairly simplistic empirical tests. Specific questions/requests for the authors: 1) Should the bounds [a,b] for the central interval s_c be given as (fixed) input to Algorithm 2? It seems like they are fixed in the algorithm, but are not provided as input or elsewhere. 2) How much the assumed bounds for s_c affect the results? Since the bounds affect the sensitivity, and thus the DP noise variance in the Laplace mechanism, the results can be made arbitrarily bad by choosing very poor bounds. Even if the bounds can be estimated under privacy (as suggested) and hence are unlikely to be totally off, the estimate is going to be noisy. It seems important to know how much this affects the reported results. Currently, I can't seem to find the bounds used in the actual tests reported in the paper anywhere, nor can I find any hint of how they were chosen. Additionally, it would be nice to have at least some test which shows the performance degradation with sub-optimal bounds. 3) On lines 247-48 you say "OPS suffices as a simpler straw man comparison". Do you think Private MCMC sampling would do better than OPS, or why is OPS a straw man? And if so, do you think you should then also compare to the private MCMC in addition to OPS? Some minor comments/questions: i) On line 36 you define calibration as being able to efficiently estimate p(theta|y). I would think calibration refers to something closer to the idea presented in the evaluation section (starting from line 252). This could be clarified. ii) On line 61 you say your "contribution is a simple release mechanism A" etc. Since the release mechanism is simply adding Laplace noise to the sufficient statistics, i.e., doing the standard stuff (see lines 111-12), I don't think this should count as a contribution, or did I misinterpret something? iii) On line 226 you mention that s_c needs to be drawn afresh for updating sigma^2. What happens if you use the previously drawn value used in s=s_l+s_c+s_u? iv) I hope you release the code for the paper (at least after the paper is published).